# Can Mean Field Control (MFC) Approximate Cooperative Multi Agent Reinforcement Learning (MARL) with Non-Uniform Interaction?

**Washim Uddin Mondal**[1, 2]        **Vaneet Aggarwal**[1]        **Satish V. Ukkusuri**[2]

[1]School of Industrial Engineering, Purdue University, West Lafayette, Indiana, USA 47907
[2]Lyles School of Civil Engineering, Purdue University, West Lafayette, Indiana, USA 47907

## Abstract

Mean-Field Control (MFC) is a powerful tool to solve Multi-Agent Reinforcement Learning (MARL) problems. Recent studies have shown that MFC can well-approximate MARL when the population size is large and the agents are exchangeable. Unfortunately, the presumption of exchangeability implies that all agents uniformly interact with one another which is not true in many practical scenarios. In this article, we relax the assumption of exchangeability and model the interaction between agents via an arbitrary doubly stochastic matrix. As a result, in our framework, the mean-field 'seen' by different agents are different. We prove that, if the reward of each agent is an affine function of the mean-field seen by that agent, then one can approximate such a non-uniform MARL problem via its associated MFC problem within an error of $e = \mathcal{O}(\frac{1}{\sqrt{N}}[\sqrt{|\mathcal{X}|} + \sqrt{|\mathcal{U}|}])$ where $N$ is the population size and $|\mathcal{X}|$, $|\mathcal{U}|$ are the sizes of state and action spaces respectively. Finally, we develop a Natural Policy Gradient (NPG) algorithm that can provide a solution to the non-uniform MARL with an error $\mathcal{O}(\max\{e, \epsilon\})$ and a sample complexity of $\mathcal{O}(\epsilon^{-3})$ for any $\epsilon > 0$.

## 1   INTRODUCTION

Multi-Agent Systems (MAS) are ubiquitous in the modern world. Many engineered systems such as transportation networks, power distribution and wireless communication systems can be modeled as MAS. Modeling, analysis and control of such systems to improve the overall performance is a central goal of research across multiple disciplines. Multi-Agent Reinforcement Learning (MARL) is a popular approach to achieve that target. In this article, we primarily focus on *cooperative* MARL where the goal is to determine *policies* for each individual agent such that the aggregate cumulative *reward* of the entire population is maximized. However, the sizes of joint state, and action spaces of the population grows exponentially with the number of agents. This makes the computation of the solution prohibitively hard for large MAS.

Two major computationally efficient approaches have been developed to tackle this problem. The first approach restricts its attention to local policies. In other words, it is assumed that each individual agent makes its decision solely based on its local state/observation. Algorithms that fall into this category are independent Q-learning (IQL) [Tan, 1993], centralised training and decentralised execution (CTDE) based algorithms such as VDN [Sunehag et al., 2017], QMIX [Rashid et al., 2018], WQMIX [Rashid et al., 2020], etc. Unfortunately, none of these algorithms can provide theoretical convergence guarantees. The other approach is called mean-field control (MFC) [Angiuli et al., 2022]. It is grounded on the idea that in an infinite population of homogeneous agents, it is sufficient to study the behaviour of only one representative agent in order to draw accurate conclusions about the whole population. Recent studies have shown that, if the agents are exchangeable, then MFC can be proven to be a good approximation of MARL [Gu et al., 2021].

Unfortunately, the idea of exchangeability essentially states that all agents in a population uniformly interact with each other (uniform means that all pairwise interactions are the same). This is not true in many practical scenarios. For example, in a traffic control network, the congestion at an intersection is highly influenced by the control policies adopted at its immediate neighbouring intersections. Moreover, the influence of an intersection on another intersection rapidly diminishes with increase of their separation distance. Non-uniform interaction is a hallmark characteristic of many other MASs such as social networks, wireless networks etc. In the absence of uniformity of the interaction between the agents, the framework of MFC no longer applies, and the problem becomes challenging. In this paper, we come up a new result which assures that even with non-uniform interac-

*Accepted for the 38th Conference on Uncertainty in Artificial Intelligence* (UAI 2022).

tions, MFC is a good choice for approximating MARL if the reward of each agent is an affine function of the mean-field distributions 'seen' by that agent. We note that the behaviour of agents in multitude of social and economic networks can be modeled via affine rewards (refer the examples given in [Chen et al., 2021]), and thus for many cases of practical interest, MFC can approximate MARL with non-uniform interactions.

## 1.1 CONTRIBUTIONS

We consider a non-uniform MARL setup where the pairwise interaction between the agents is described by an arbitrary doubly stochastic matrix (DSM). As a result of non-uniform interaction, the so-called mean-field effect of the population on an agent is determined by the identity of the agent. This is in stark contrast with other existing works [Gu et al., 2021, Mondal et al., 2022] where the presumption of exchangeability washes away the dependence on identity. We demonstrate that, if the reward of each agent is an affine function of the mean-field distribution 'seen' by that agent, then the standard MFC approach can approximate the non-uniform MARL with an error bound of $e \triangleq \mathcal{O}(\frac{1}{\sqrt{N}}[\sqrt{|\mathcal{X}|} + \sqrt{|\mathcal{U}|}])$, where $N$ is the number of agents and $|\mathcal{X}|, |\mathcal{U}|$ indicate the sizes of state and action spaces of individual agent.

We would like to emphasize the importance of this result. MFC is traditionally seen as an approximation method of MARL when the agents are exchangeable and hence their interactions are uniform. Uniformity allows us to solve MFC problems by tracking only one representative agent. In this paper, we show that, under certain conditions, a non-uniform MARL can also be approximated by the MFC approach. Thus, although the non-uniform interaction is a major part of the original MARL problem, the assumed affine structure of the reward function allows us to evade non-uniformity while obtaining an approximate solution. The key result is established in Lemma B.7 (Appendix B.2) where, using the affine structure of the reward function, we show that the instantaneous reward generated from non-uniform MARL can be closely approximated by MFC-generated instantaneous reward.

Finally, using the results of [Liu et al., 2020], in section 5, we design a natural policy-gradient based algorithm that can solve MFC within an error of $\mathcal{O}(\epsilon)$ for any $\epsilon > 0$, with a sample complexity of $\mathcal{O}(\epsilon^{-3})$. Invoking our approximation result, we prove that the devised algorithm can yield a solution that is $\mathcal{O}(\max\{e, \epsilon\})$ error away from the optimal MARL solution, with a sample complexity of $\mathcal{O}(\epsilon^{-3})$ for any $\epsilon > 0$.

## 1.2 RELATED WORKS

**Single Agent RL:** The classical algorithms in single agent learning include tabular Q-learning Watkins and Dayan [1992], SARSA Rummery and Niranjan [1994], etc. Although they provide theoretical guarantees, these algorithms can only be applied to small state-action space based systems due to their large memory requirements. Recently Neural Network (NN) based $Q$-iteration Mnih et al. [2015], and policy gradient Mnih et al. [2016] algorithms have becomes popular due to the large expressive power of NN. However, they cannot be applied to large MAS due to the exponential blow-up of joint state-space.

**MFC as an Approximation to Uniform MARL:** Recently, MFC is gaining traction as a scalable approximate solution to uniform MARL. On the theory side, recently it has been proven that MFC can approximate uniform MARL within an error of $\mathcal{O}(1/\sqrt{N})$ [Gu et al., 2021]. However, the result relies on the assumption that all agents are homogeneous. Later, this approximation result was extended to heterogeneous agents [Mondal et al., 2022]. We would like to clarify that the idea of heterogeneity is different from the idea of non-uniformity. In the first case, the agents are divided into multiple classes. However, the identities of different agents within a given class are irrelevant. In contrast, non-uniform interaction takes the identity of each agent into account.

**Graphon Approximation:** One possible approach to consider non-uniform agent interaction is the notion of Graphon mean-field, which is recently gaining popularity in the *non-cooperative* MARL setup [Caines and Huang, 2019, Cui and Koeppl, 2021]. The main idea is to approximate the finite indices of the agents as a continuum of real numbers and the discrete interaction graph between agents as a continuous, symmetric, measurable function, called graphon, in the asymptotic limit of infinite population. The unfortunate consequence of this approximation is that one is left to deal with an infinite dimensional mean-field distribution. In order to obtain practical solution from graphon-approximation, one must therefore discretise the continuum of agent indices [Cui and Koeppl, 2021], which limits the use of this approximation. Our paper establishes that for affine reward functions, we do not need to go to the complexity of Graphon approximation.

**Applications of MFC:** Alongside the theory, MFC has also become popular as an application tool. It has been used in ride-sharing [Al-Abbasi et al., 2019], epidemic management [Watkins et al., 2016], congestion control in road network [Wang et al., 2020] etc.

**Learning Algorithms for MFC:** Both model-free [Angiuli et al., 2022, Gu et al., 2021] and model-based [Pasztor et al., 2021] Q-learning algorithms have been proposed in the literature to solve uniform MARL via MFC with homogeneous agents. Recently, [Mondal et al., 2022] proposed a policy-gradient algorithm for heterogeneous-MFC.

# 2 COOPERATIVE MARL WITH NON-UNIFORM INTERACTION

We consider a system comprising of $N$ interacting agents. The (finite) state and action spaces of each agent are denoted as $\mathcal{X}$, and $\mathcal{U}$ respectively. Time is assumed to belong to the discrete set, $\mathbb{T} \triangleq \{0, 1, 2, \cdots\}$. The state and action of $i$-th agent at time $t$ are symbolized as $x_t^i$ and $u_t^i$. The empirical state and action distributions of the population of agents at time $t$ are denoted by $\boldsymbol{\mu}_t^N$, and $\boldsymbol{\nu}_t^N$ respectively, and defined as follows.

$$\boldsymbol{\mu}_t^N(x) \triangleq \frac{1}{N} \sum_{i=1}^N \delta(x_t^i = x), \; \forall x \in \mathcal{X}, \forall t \in \mathbb{T} \quad (1)$$

$$\boldsymbol{\nu}_t^N(u) \triangleq \frac{1}{N} \sum_{i=1}^N \delta(u_t^i = u), \; \forall u \in \mathcal{U}, \forall t \in \mathbb{T} \quad (2)$$

where $\delta(\cdot)$ is the indicator function.

Each agent, $i \in [N] \triangleq \{1, \cdots, N\}$ is endowed with a reward function $r$ and a state transition function $P$ that are of the following forms: $r : \mathcal{X} \times \mathcal{U} \times \mathcal{P}(\mathcal{X}) \times \mathcal{P}(\mathcal{U}) \to \mathbb{R}$ and $P : \mathcal{X} \times \mathcal{U} \times \mathcal{P}(\mathcal{X}) \times \mathcal{P}(\mathcal{U}) \to \mathcal{P}(\mathcal{X})$ where $\mathcal{P}(\cdot)$ is the set of all Borel probability measures over its argument. In particular, $r, P$ take the followings as arguments: (a) the state, $x_t^i$ and action, $a_t^i$ of the corresponding agent and (b) the weighted state distribution $\boldsymbol{\mu}_t^{i,N}$ and the weighted action distribution $\boldsymbol{\nu}_t^{i,N}$ of the population as seen from the perspective of the agent. The terms $\boldsymbol{\mu}_t^{i,N}$ and $\boldsymbol{\nu}_t^{i,N}$ are defined as follows.

$$\boldsymbol{\mu}_t^{i,N}(x) \triangleq \sum_{j=1}^N W(i,j)\delta(x_t^j = x), \; \forall x \in \mathcal{X}, \forall t \in \mathbb{T} \quad (3)$$

$$\boldsymbol{\nu}_t^{i,N}(u) \triangleq \sum_{j=1}^N W(i,j)\delta(u_t^j = u), \; \forall u \in \mathcal{U}, \forall t \in \mathbb{T} \quad (4)$$

The function $W : [N] \times [N] \to [0, 1]$ dictates the influence of one agent on another. In particular, $W(i,j)$ specifies how $j$-th agent influences $i$-th agent's reward and transition functions. Observe that, for $\boldsymbol{\mu}_t^{i,N}$, and $\boldsymbol{\nu}_t^{i,N}$ to be probability distributions, $W$ must be right-stochastic i.e.,

$$\sum_{j=1}^N W(i,j) = 1, \; \forall i \in \{1, \cdots, N\} \quad (5)$$

In summary, the reward received by the $i$-th agent at time $t$ can be expressed as $r(x_t^i, u_t^i, \boldsymbol{\mu}_t^{i,N}, \boldsymbol{\nu}_t^{i,N})$. Moreover, the state of the agent at time $t + 1$ is decided by the following probability law: $x_{t+1}^i \sim P(x_t^i, u_t^i, \boldsymbol{\mu}_t^{i,N}, \boldsymbol{\nu}_t^{i,N})$. We would like to point out that, in contrast to our framework, existing works assume reward and state transition to be functions of $\boldsymbol{\mu}_t^N, \boldsymbol{\nu}_t^N$, thereby making the influence of population to be identical for every agent [Mondal et al., 2022, Gu et al., 2021]. If we take the influence function $W$ to be uniform i.e.,

$W(i,j) = 1/N, \forall i, j \in [N]$, then $\forall i \in [N]$, $\boldsymbol{\mu}_t^{i,N} = \boldsymbol{\mu}_t^N$, and $\boldsymbol{\nu}_t^{i,N} = \boldsymbol{\nu}_t^N$, which forces our framework to collapse onto that described in the above mentioned papers.

At time $t \in \mathbb{T}$, each agent is also presumed to have a policy function $\pi_t : \mathcal{X} \times \mathcal{P}(\mathcal{X}) \to \mathcal{P}(\mathcal{U})$ that maps $(x_t^i, \boldsymbol{\mu}_t^{i,N})$ to a distribution over the action space, $\mathcal{U}$. In simple words, a policy function $\pi_t$ is a rule that (probabilistically) dictates what action must be chosen by an agent given its current state and the mean-distribution of the population as observed by the agent. Note that the policy function is presumed to be the same for all the agents as the reward function, $r$ and the transition function, $P$ is taken to homogeneous across the population. Homogeneity of $r, P$ is a common assumption in the mean-field literature [Gu et al., 2021, Vasal et al., 2021].

For a given set of initial states $\boldsymbol{x}_0 \triangleq \{x_0^i\}_{i \in \mathbb{N}}$, the value of the sequence of policies, $\boldsymbol{\pi} \triangleq \{\pi_t\}_{t \in \mathbb{T}}$, for the $i$-th agent is defined as follows.

$$v_i(\boldsymbol{x}_0, \boldsymbol{\pi}) \triangleq \sum_{t \in \mathbb{T}} \gamma^t \mathbb{E}\left[ r\left(x_t^i, u_t^i, \boldsymbol{\mu}_t^{i,N}, \boldsymbol{\nu}_t^{i,N}\right)\right] \quad (6)$$

where $\boldsymbol{\mu}_t^{i,N}, \boldsymbol{\nu}_t^{i,N}$ are defined by $(3), (4)$, respectively, and the expectation is computed over all the state-action trajectories generated by the transition function $P$ and the sequence of policy functions, $\boldsymbol{\pi}$. The term, $\gamma \in [0, 1]$ is called the time discount factor. We would like to emphasize that the value function $v_i$ is dependent on the interaction matrix $W$ (because so are $\boldsymbol{\mu}_t^{i,N}$, and $\boldsymbol{\nu}_t^{i,N}$). However, such dependence is not explicitly shown to keep the notation uncluttered. The average value function of the entire population is expressed as below.

$$v_{\text{MARL}}(\boldsymbol{x}_0, \boldsymbol{\pi}) = \frac{1}{N} \sum_{i=1}^N v_i(\boldsymbol{x}_0, \boldsymbol{\pi}) \quad (7)$$

The goal of MARL is to maximize $v_{\text{MARL}}(\boldsymbol{x}_0, .)$ over all policy sequences $\boldsymbol{\pi}$. Such an optimization is hard to solve in general, especially for large $N$.

Before concluding this section, we would like to point out two important observations that will be extensively used in many of our forthcoming results.

**Remark 1.** $\forall t \in \mathbb{T}$, the random variables $\{u_t^i\}_{i \in [N]}$ are conditionally independent given $\{x_t^i\}_{i \in [N]}$. In other words, given current states, each agent chooses its action independent of each other.

**Remark 2.** $\forall t \in \mathbb{T}$, the random variables $\{x_{t+1}^i\}_{i \in [N]}$ are conditionally independent given $\{x_t^i\}_{i \in [N]}$, and $\{u_t^i\}_{i \in [N]}$. In other words, given current states and actions, the next state of each agent evolves independent of each other.

## 3 MEAN-FIELD CONTROL

MFC is an approximation method of $N-$agent MARL that takes away many of the complexities of the later. The main idea of MFC is to consider an infinite population of homogeneous agents, instead of a finite population as considered in MARL. The advantage of such presumption is that it allows us to draw accurate inferences about the whole population by tracking only a single representative agent. Unfortunately, as stated before, such approximation method is known to work [Gu et al., 2021] when the interactions between different agents are uniform, i.e., $W(i, j) = 1/N, \forall i, j \in [N]$. In this article, we shall show that, under certain conditions, we can show MFC as an approximation of MARL, even with non-uniform $W$. Below we describe the MFC method.

As explained above, in MFC, we only need to track a single representative agent. Let the state and action of the agent at time $t$ be denoted as $x_t$, and $u_t$ respectively. Also, let $\boldsymbol{\mu}_t$, $\boldsymbol{\nu}_t$ be the state, and action distributions of the infinite population at time $t$. The reward and state transition laws of the representative at time $t$ are denoted as $r(x_t, u_t, \boldsymbol{\mu}_t, \boldsymbol{\nu}_t)$ and $P(x_t, u_t, \boldsymbol{\mu}_t, \boldsymbol{\nu}_t)$, respectively. For a given policy sequence $\boldsymbol{\pi} \triangleq \{\pi_t\}_{t \in \mathbb{T}}$, the action distribution $\boldsymbol{\nu}_t$ can be expressed as a deterministic function of the state distribution, $\boldsymbol{\mu}_t$ as follows.

$$\boldsymbol{\nu}_t = \nu^{\mathrm{MF}}(\boldsymbol{\mu}_t, \pi_t) \triangleq \sum_{x \in \mathcal{X}} \pi_t(x, \boldsymbol{\mu}_t)\boldsymbol{\mu}_t(x) \qquad (8)$$

In a similar fashion, the state distribution at time $t + 1$ can be written as a deterministic function of $\boldsymbol{\mu}_t$ as follows.

$$\begin{aligned}
\boldsymbol{\mu}_{t+1} &= P^{\mathrm{MF}}(\boldsymbol{\mu}_t, \pi_t) \\
&\triangleq \sum_{x \in \mathcal{X}} \sum_{u \in \mathcal{U}} P(x, u, \boldsymbol{\mu}_t, \nu^{\mathrm{MF}}(x, \boldsymbol{\mu}_t)) \\
&\qquad \times \pi_t(x, \boldsymbol{\mu}_t)(u)\boldsymbol{\mu}_t(x)
\end{aligned} \qquad (9)$$

For an initial state distribution $\boldsymbol{\mu}_0$, the value of a sequence of policies $\boldsymbol{\pi} \triangleq \{\pi_t\}_{t \in \mathbb{T}}$, is defined as written below.

$$\begin{aligned}
v_{\mathrm{MF}}(\boldsymbol{\mu}_0, \boldsymbol{\pi}) &\triangleq \sum_{t \in \mathbb{T}} \gamma^t r^{\mathrm{MF}}(\boldsymbol{\mu}_t, \pi_t), \\
\text{where } r^{\mathrm{MF}}(\boldsymbol{\mu}_t, \pi_t) &\triangleq \sum_{x \in \mathcal{X}} \sum_{u \in \mathcal{U}} r(x, u, \boldsymbol{\mu}_t, \nu^{\mathrm{MF}}(\boldsymbol{\mu}_t, \pi_t)) \\
&\qquad \times \pi_t(x, \boldsymbol{\mu}_t)(u)\boldsymbol{\mu}_t(x)
\end{aligned} \qquad (10)$$

The term $r^{\mathrm{MF}}(\boldsymbol{\mu}_t, \pi_t)$ indicates the average reward of the population. Alternatively, it can also be expressed as the ensemble average of the reward of the representative agent i.e., $r^{\mathrm{MF}}(\boldsymbol{\mu}_t, \pi_t) = \mathbb{E}[r(x_t, u_t, \boldsymbol{\mu}_t, \boldsymbol{\nu}_t)]$ where the expectation is computed over all possible states $x_t \sim \boldsymbol{\mu}_t$, and actions $u_t \sim \pi_t(x_t, \boldsymbol{\mu}_t)$ at time $t$. The mean distributions $\boldsymbol{\mu}_t, \boldsymbol{\nu}_t$ are sequentially determined by $(8), (9)$ from a given initial state distribution, $\boldsymbol{\mu}_0$.

The goal of MFC is to maximize $v_{\mathrm{MF}}(\boldsymbol{\mu}_0, \cdot)$ over all policy sequences. In the next section, we shall demonstrate that, under certain conditions, $v_{\mathrm{MARL}}$ is well-approximated by $v_{\mathrm{MF}}$. Therefore, in order to solve MARL, it is sufficient to solve its associated MFC.

It is worthwhile to point out that $\boldsymbol{\mu}_t, \boldsymbol{\nu}_t$ can be thought of as limiting values of the empirical distributions $\boldsymbol{\mu}_t^N, \boldsymbol{\nu}_t^N$ in the asymptotic limit of infinite population. Note that, $\boldsymbol{\mu}_t^N, \boldsymbol{\nu}_t^N$ and thereby, $\boldsymbol{\mu}_t, \boldsymbol{\nu}_t$ are NOT dependent on $W$. This makes the MFC problem agnostic of $W$. In contrary, agents in the $N-$agent MARL problem are influenced by the weighted mean-field distribution $\{\boldsymbol{\mu}_t^{i,N}, \boldsymbol{\nu}_t^{i,N}\}_{i \in [N]}$ which do depend on $W$ via $(3), (4)$. Therefore, unlike in the existing works, the mean-field representative in our case cannot be described as a randomly chosen typical agent in the limit $N \to \infty$. The concept of mean-field representative, in our work, is a useful construct that, under certain conditions, can provide well-approximated solution to MARL.

In the next section, we describe how these seemingly incompatible frameworks, namely non-uniform MARL where the behaviour of agents are dependent on $W$, and the framework of $W$-agnostic MFC, can be merged together.

## 4 MFC AS AN APPROXIMATION TO NON-UNIFORM MARL

Before formally stating our main result, we would like to describe the assumptions that the result is grounded upon. Our first assumption is on the structure of state-transition function.

**Assumption 1.** *The state-transition function $P$ is Lipschitz continuous with parameter $L_P$ with respect to the mean-distribution arguments. Mathematically, the inequality,*

$$\begin{aligned}
|P(x, u, &\boldsymbol{\mu}_1, \boldsymbol{\nu}_1) - P(x, u, \boldsymbol{\mu}_2, \boldsymbol{\nu}_2)|_1 \\
&\leq L_P[|\boldsymbol{\mu}_1 - \boldsymbol{\mu}_2|_1 + |\boldsymbol{\nu}_1 - \boldsymbol{\nu}_2|_1]
\end{aligned}$$

*holds $\forall x \in \mathcal{X}, \forall u \in \mathcal{U}, \forall \boldsymbol{\mu}_1, \boldsymbol{\mu}_2 \in \mathcal{P}(\mathcal{X})$ and $\forall \boldsymbol{\nu}_1, \boldsymbol{\nu}_2 \in \mathcal{P}(\mathcal{U})$. The symbol $|\cdot|_1$ denotes $L_1$ norm.*

Assumption 1 states that the transition function, $P$, is Lipschitz continuous with respect to its mean-field arguments. Essentially, this implies that if the state-distribution changes from $\boldsymbol{\mu}$ to $\boldsymbol{\mu} + \Delta\boldsymbol{\mu}$, then the corresponding change in the transition-function can be bounded by a term proportional to $|\Delta\boldsymbol{\mu}|_1$. Similar property holds for the change in the action-distribution. This useful assumption commonly appears in the mean-field literature [Gu et al., 2021, Mondal et al., 2022, Carmona et al., 2018].

The second assumption is on the structure of $r$, the reward function.

**Assumption 2.** *The reward function, $r$ is affine with respect to mean-distribution arguments. Mathematically, for some*

$\boldsymbol{a} \in \mathbb{R}^{|\mathcal{X}|}, \boldsymbol{b} \in \mathbb{R}^{|\mathcal{U}|}$, and $f : \mathcal{X} \times \mathcal{U} \to \mathbb{R}$, the equality,

$$r(x, u, \boldsymbol{\mu}, \boldsymbol{\nu}) = \boldsymbol{a}^T \boldsymbol{\mu} + \boldsymbol{b}^T \boldsymbol{\nu} + f(x, u)$$

holds $\forall x \in \mathcal{X}$, $\forall u \in \mathcal{U}$, $\forall \boldsymbol{\mu} \in \mathcal{P}(\mathcal{X})$, and $\forall \boldsymbol{\nu} \in \mathcal{P}(\mathcal{U})$.

Assumption 2 dictates that the reward is an affine function of the mean-field distributions. Although this assumption does not allow us to encapsulate a large variety of reward functions, we would like to point out that the behaviour of agents in multitude of social and economic networks can be modeled via affine rewards (refer the examples given in [Chen et al., 2021]). We shall provide one explicit example at the end of this section. We would also like to reiterate that the benefit of this seemingly restrictive assumption of affine reward is it allows us to apply the principles of MFC to an arbitrarily interacting $N$-agent system which is notoriously complex to solve in general.

The immediate corollary of Assumption 2 is that the reward function is bounded and Lipschitz continuous. The formal proposition is given below.

**Corollary 1.** *If the reward function, $r$ satisfies Assumption 2, then for some $M_R, L_R > 0$, the following holds*

$$(a) |r(x, u, \boldsymbol{\mu}_1, \boldsymbol{\nu}_1)| \leq M_R,$$
$$(b) |r(x, u, \boldsymbol{\mu}_1, \boldsymbol{\nu}_1) - r(x, u, \boldsymbol{\mu}_2, \boldsymbol{\nu}_2)|$$
$$\leq L_R \left[|\boldsymbol{\mu}_1 - \boldsymbol{\mu}_2|_1 + |\boldsymbol{\nu}_1 - \boldsymbol{\nu}_2|_1\right]$$

$\forall x \in \mathcal{X}$, $\forall u \in \mathcal{U}$, $\forall \boldsymbol{\mu}_1, \boldsymbol{\mu}_2 \in \mathcal{P}(\mathcal{X})$, and $\forall \boldsymbol{\nu}_1, \boldsymbol{\nu}_2 \in \mathcal{P}(\mathcal{U})$.

The third assumption concerns the set of allowable policy functions.

**Assumption 3.** *The set of allowable policy functions, $\Pi$, is such that each of its element is Lipschitz continuous with respect to its mean-state distribution argument. Mathematically, $\forall \pi \in \Pi$, the following inequality holds*

$$|\pi(x, \boldsymbol{\mu}_1) - \pi(x, \boldsymbol{\mu}_2)|_1 \leq L_Q |\boldsymbol{\mu}_1 - \boldsymbol{\mu}_2|_1$$

*for some $L_Q > 0$ and $\forall x \in \mathcal{X}$, $\forall \boldsymbol{\mu}_1, \boldsymbol{\mu}_2 \in \mathcal{P}(\mathcal{X})$.*

Assumption 3 states that the allowable policy functions must be Lipschitz continuous with respect to its state-distribution argument. Such requirement typically holds for neural network based policies and are commonly presumed to be true in the literature [Gu et al., 2021, Cui and Koeppl, 2021, Pasztor et al., 2021].

The final assumption imposes some constraints on the interaction function, $W$.

**Assumption 4.** *The interaction function, $W$ is such that,*

$$\sum_{i=1}^{N} W(i, j) = 1, \ \forall j \in \{1, \cdots, N\} \quad (11)$$

*In conjunction with (5), this assumption implies that $W$ is doubly-stochastic.*

Assumption 4 requires $W$ to be an $N \times N$ doubly stochastic matrix (DSM). Such presumption is commonly applied in many multi-agent tasks, e.g., distributed consensus [Alaviani and Elia, 2019a], distributed optimization [Alaviani and Elia, 2019b], and multi-agent learning [Wai et al., 2018].

We now state our main result.

**Theorem 1.** *Let, $\boldsymbol{x}_0 \triangleq \{x_0^i\}_{i \in [N]}$ be the initial states in an $N$-agent non-uniform MARL problem and $\boldsymbol{\mu}_0$ be its associated empirical distribution defined by $(1)$. Assume $\Pi$ to be a set of policies that obeys Assumption 3, and $\boldsymbol{\pi} \triangleq \{\pi_t\}_{t \in \mathbb{T}}$ is a sequence of policies such that $\pi_t \in \Pi$, $\forall t \in \mathbb{T}$. If Assumption 1, 2 and 4 hold, then*

$$|v_{\mathrm{MARL}}(\boldsymbol{x}_0, \boldsymbol{\pi}) - v_{\mathrm{MF}}(\boldsymbol{\mu}_0, \boldsymbol{\pi})| \leq C_R \frac{\sqrt{|\mathcal{U}|}}{\sqrt{N}} \frac{1}{1 - \gamma}$$
$$+ \frac{1}{\sqrt{N}} \left[\sqrt{|\mathcal{X}|} + \sqrt{|\mathcal{U}|}\right] \frac{S_R C_P}{S_P - 1} \left[\frac{1}{1 - \gamma S_P} - \frac{1}{1 - \gamma}\right] \quad (12)$$

*whenever $\gamma S_P < 1$ where $S_P \triangleq (1 + L_Q) + L_P(2 + L_Q)$, $S_R \triangleq M_R(1 + L_Q) + L_R(2 + L_Q)$, $C_P \triangleq 2 + L_P$, and $C_R \triangleq |\boldsymbol{b}|_1 + M_F$. The parameters $L_P, \boldsymbol{b}, L_Q, L_R, M_R$ have been defined in Assumption 1, 2, 3, and Corollary 1, respectively. The term $M_F$ is such that $|f(x, u)| \leq M_F$, $\forall x \in \mathcal{X}$, $\forall u \in \mathcal{U}$ where $f$ is stated in Assumption 2. The functions $v_{\mathrm{MARL}}$, and $v_{\mathrm{MF}}$ are defined in $(7), (10)$ respectively.*

Theorem 1 has an important implication. Specifically, it states that, if reward and transition functions respectively are affine and Lipschitz continuous functions of the mean-distributions, and the interaction between the agents is described by a DSM, then the solution of MFC is at most $\mathcal{O}(1/\sqrt{N})$ error away from the solution of the non-uniform MARL problem. Therefore, the larger the number of agents, the better is the MFC-based approximation. It also describes how the approximation error changes with the sizes of the state, and action spaces. Specifically, if all other parameters are kept fixed, then the error increases as $\mathcal{O}(\sqrt{|\mathcal{X}|} + \sqrt{|\mathcal{U}|})$. In other words, if individual state and action spaces are large, then MFC may not be a good approximation to non-uniform MARL.

Now we shall discuss one example where the reward, transition function and the interaction function satisfy Assumption 1, 2, and 4 respectively.

**Example 1.** *A version of this model has been adapted in [Subramanian and Mahajan, 2019] and [Chen et al., 2021]. Consider a network of $N$ firms operated by a single operator. All of the firms produce the same product but with varying quality. A discrete set $\mathcal{X} \triangleq \{1, 2, \cdots, Q\}$ (state-space) describes the possible levels of quality of the product. At each time instant, each firm decides whether to invest to improve the quality of its product which leads to the following action set: $\mathcal{U} = \{0, 1\}$. If at time $t$, the $i$-th firm decides to invest,*

*i.e., $u_t^i = 1$, its current quality, $x_t^i$, improves according to the following transition law.*

$$x_{t+1}^i = \begin{cases} x_t^i + \left\lfloor \chi \left(1 - \frac{\bar{\boldsymbol{\mu}}_t^{i,N}}{Q}\right)(Q - x_t^i) \right\rfloor & \text{if } u_t^i = 1, \\ x_t^i & \text{otherwise} \end{cases}$$

*where $\chi$ is a uniform random variable between $[0, 1]$, and $\bar{\boldsymbol{\mu}}_t^{i,N}$ is average product quality of its $K < N$ neighbouring firms. The intuition is that improving product quality might be difficult if the quality maintained in the local economy is high. Formally, we assume that each firm equally influences and is influenced by $K$ other firms. Hence, $W(i, j) = 1/K$ for all $i, j \in [N]$ that influence each other and $W(i, j) = 0$ otherwise. The local average product quality is computed as, $\bar{\boldsymbol{\mu}}_t^{i,N} \triangleq \sum_{x \in \mathcal{X}} x \boldsymbol{\mu}_t^{i,N}(x)$ where $\boldsymbol{\mu}_t^{i,N}$ is given in (3). At time $t$, the $i$-th firm earns a positive reward, $\alpha_R x_t^i$ due to its revenue, a negative reward, $\beta_R \bar{\boldsymbol{\mu}}_t^{i,N}$ due to the average local quality, and a cost $\lambda_R u_t^i$ due to investment. Hence, the total reward can be expressed as follows.*

$$r(x_t^i, u_t^i, \boldsymbol{\mu}_t^{i,N}, \boldsymbol{\nu}_t^{i,N}) = \alpha_R x_t^i - \beta_R \bar{\boldsymbol{\mu}}_t^{i,N} - \lambda_R u_t^i$$

*Clearly, in this example, Assumption 1, 2, and 4 are satisfied.*

## 4.1 PROOF OUTLINE

In this subsection, we shall provide a brief sketch of the proof of Theorem 1.

*Step 0*: The difference between $v_{\text{MARL}}$ and $v_{\text{MF}}$ is essentially the time-discounted sum of differences between the average $N$-agent reward and average mean-field (MF) reward at time $t$. Our first goal, therefore, is to estimate the difference between these rewards.

*Step 1*: Average $N$-agent reward at $t$ depends on weighted empirical distributions $\{\boldsymbol{\mu}_t^{i,N}\}_{i \in [N]}$, $\{\boldsymbol{\nu}_t^{i,N}\}_{i \in [N]}$ whereas average MF reward depends on the distributions $\boldsymbol{\mu}_t, \boldsymbol{\nu}_t$. To estimate their difference, we first compute the difference between average $N$-agent reward at $t$ and average MF reward at the same instant generated from the distribution $\boldsymbol{\mu}_t^N$. This estimate is provided by Lemma B.7 in the Appendix. Assumption 2 is invoked to establish this result.

*Step 2*: Next we estimate the difference between the average MF reward generated by $\boldsymbol{\mu}_t^N$ and that generated by $\boldsymbol{\mu}_t$. Lemma B.3 in the Appendix bounds this difference by a term proportional to $|\boldsymbol{\mu}_t^N - \boldsymbol{\mu}_t|$.

*Step 3*: Using Lemma B.2 and B.6, we now establish a recursive relation on $|\boldsymbol{\mu}_t^N - \boldsymbol{\mu}_t|$. Via induction, we can now write this difference as a function of $t$.

*Step 4*: Finally, by computing a time-discounted sum of all the upper bounds described above, we arrive at the desired result.

# 5 SOLUTION OF MFC VIA NATURAL POLICY GRADIENT ALGORITHM

In this section, we develop a Natural Policy Gradient (NPG) algorithm to solve the MFC problem. By virtue of Theorem 1, it provides an approximate solution to the non-uniform MARL problem. Recall from section 3 that, in MFC, it is sufficient to track only one representative agent. At time $t$, that agent takes its decision $u_t$ based on its own state $x_t$, and the mean-field state distribution $\boldsymbol{\mu}_t$. Thus, MFC essentially reduces to a single-agent Markov Decision Problem (MDP) with extended state space $\mathcal{X} \times \mathcal{P}(\mathcal{X})$ and action space $\mathcal{U}$. To solve MFC, it is therefore sufficient to consider only stationary policies [Puterman, 2014].

Let the set of stationary policies be denoted by $\Pi$ and its elements be parameterized by $\Phi \in \mathbb{R}^d$. For a given policy $\pi_\Phi \in \Pi$, we shall define its sequence as $\boldsymbol{\pi}_\Phi \triangleq \{\pi_\Phi, \pi_\Phi, \cdots\}$. Let, $Q_\Phi$ be the Q-function associated with policy $\pi_\Phi$. We define $Q_\Phi(x, \boldsymbol{\mu}, u)$ for arbitrary $x \in \mathcal{X}$, $\boldsymbol{\mu} \in \mathcal{P}(\mathcal{X})$, and $u \in \mathcal{U}$, as follows.

$$Q_\Phi(x, \boldsymbol{\mu}, u) \triangleq$$
$$\mathbb{E}\left[\sum_{t=0}^{\infty} \gamma^t r(x_t, u_t, \boldsymbol{\mu}_t, \boldsymbol{\nu}_t) \Big| x_0 = x, \boldsymbol{\mu}_0 = \boldsymbol{\mu}, u_0 = u\right] \tag{13}$$

where the expectation is over $u_{t+1} \sim \pi_\Phi(x_{t+1}, \boldsymbol{\mu}_{t+1})$, and $x_{t+1} \sim P(x_t, u_t, \boldsymbol{\mu}_t, \boldsymbol{\nu}_t)$, $\forall t \in \mathbb{T}$. The mean-field distributions $\{\boldsymbol{\mu}_{t+1}, \boldsymbol{\nu}_t\}_{t \in \mathbb{T}}$ are updated via deterministic update equations (8), and (9). We now define the advantage function as follows.

$$A_\Phi(x, \boldsymbol{\mu}, u) \triangleq Q_\Phi(x, \boldsymbol{\mu}, u) - \mathbb{E}[Q_\Phi(x, \boldsymbol{\mu}, u)] \tag{14}$$

where the expectation is over $u \sim \pi_\Phi(x, \boldsymbol{\mu})$.

Let, $v_{\text{MF}}^*(\boldsymbol{\mu}_0) = \sup_{\Phi \in \mathbb{R}^d} v_{\text{MF}}(\boldsymbol{\mu}_0, \boldsymbol{\pi}_\Phi)$ where $v_{\text{MF}}$ is the value function of MFC problem and is defined in (10). Let, $\{\Phi_j\}_{j=1}^J$ be a sequence of parameters that are generated by the NPG algorithm [Liu et al., 2020, Agarwal et al., 2021] as follows.

$$\Phi_{j+1} = \Phi_j + \eta \mathbf{w}_j, \mathbf{w}_j \triangleq \arg\min_{\mathbf{w} \in \mathbb{R}^d} L_{\zeta_{\boldsymbol{\mu}_0}^{\Phi_j}}(\mathbf{w}, \Phi_j) \tag{15}$$

The term $\eta$ is defined as the learning parameter. The function $L_{\zeta_{\boldsymbol{\mu}_0}^{\Phi_j}}$ and the distribution $\zeta_{\boldsymbol{\mu}_0}^{\Phi_j}$ are defined below.

$$L_{\zeta_{\boldsymbol{\mu}_0}^{\Phi'}}(\mathbf{w}, \Phi) \triangleq \mathbb{E}_{(x,\boldsymbol{\mu},u) \sim \zeta_{\boldsymbol{\mu}_0}^{\Phi'}}\left[\Big(A_\Phi(x, \boldsymbol{\mu}, u)\right.$$
$$\left. - (1-\gamma)\mathbf{w}^{\mathrm{T}} \nabla_\Phi \log \pi_\Phi(x, \boldsymbol{\mu})(u)\Big)^2\right], \tag{16}$$

$$\zeta_{\boldsymbol{\mu}_0}^{\Phi'}(x, \boldsymbol{\mu}, u) \triangleq \sum_{\tau=0}^{\infty} \gamma^\tau \mathbb{P}(x_\tau = x, \boldsymbol{\mu}_\tau = \boldsymbol{\mu}, u_\tau = u| \tag{17}$$
$$x_0 = x, \boldsymbol{\mu}_0 = \boldsymbol{\mu}, u_0 = u, \boldsymbol{\pi}_{\Phi'})(1-\gamma)$$

NPG update (15) indicates that, at each iteration, one must solve another minimization problem to obtain the gradient direction. It can be solved by applying a stochastic gradient descent (SGD) approach. In particular, the update equation, in this case, turns out to be the following: $\mathbf{w}_{j,l+1} = \mathbf{w}_{j,l} - \alpha \mathbf{h}_{j,l}$ [Liu et al., 2020]. The term $\alpha$ is the learning rate for this sub-problem. The update direction $\mathbf{h}_{j,l}$ can be defined as follows.

$$
\mathbf{h}_{j,l} \triangleq \left( \mathbf{w}_{j,l}^{\mathrm{T}} \nabla_{\Phi_j} \log \pi_{\Phi_j}(x, \boldsymbol{\mu})(u) \right.
$$
$$
\left. - \frac{1}{1-\gamma} \hat{A}_{\Phi_j}(x, \boldsymbol{\mu}, u) \right) \nabla_{\Phi_j} \log \pi_{\Phi_j}(x, \boldsymbol{\mu})(u) \tag{18}
$$

where $(x, \boldsymbol{\mu}, u) \sim \zeta_{\boldsymbol{\mu}_0}^{\Phi_j}$, and $\hat{A}_{\Phi_j}$ is a unbiased estimator of $A_{\Phi_j}$. The process to obtain the samples and the estimator has been detailed in Algorithm 1 in the Appendix I. We would like to point out that Algorithm 1 is based on Algorithm 3 of [Agarwal et al., 2021]. We summarize the whole NPG process in Algorithm 1.

---

**Algorithm 1** Natural Policy Gradient

**Input:** $\eta, \alpha$: Learning rates, $J, L$: Number of execution steps
$\mathbf{w}_0, \Phi_0$: Initial parameters, $\boldsymbol{\mu}_0$: Initial state distribution
**Initialization:** $\Phi \leftarrow \Phi_0$
1: **for** $j \in \{0, 1, \cdots, J-1\}$ **do**
2:     $\mathbf{w}_{j,0} \leftarrow \mathbf{w}_0$
3:     **for** $l \in \{0, 1, \cdots, L-1\}$ **do**
4:         Sample $(x, \boldsymbol{\mu}, u) \sim \zeta_{\boldsymbol{\mu}_0}^{\Phi_j}$ and $\hat{A}_{\Phi_j}(x, \boldsymbol{\mu}, u)$ using Algorithm 1
5:         Compute $\mathbf{h}_{j,l}$ using (18)
           $\mathbf{w}_{j,l+1} \leftarrow \mathbf{w}_{j,l} - \alpha \mathbf{h}_{j,l}$
6:     **end for**
7:     $\mathbf{w}_j \leftarrow \frac{1}{L} \sum_{l=1}^{L} \mathbf{w}_{j,l}$
8:     $\Phi_{j+1} \leftarrow \Phi_j + \eta \mathbf{w}_j$
9: **end for**
**Output:** $\{\Phi_1, \cdots, \Phi_J\}$: Policy parameters

---

The global converge of NPG is stated in Lemma 1 which is a direct consequence of Theorem 4.9 of [Liu et al., 2020]. However, the following assumptions are needed to establish the Lemma. These are similar to Assumptions 2.1, 4.2, 4.4 respectively in [Liu et al., 2020].

**Assumption 5.** $\forall \Phi \in \mathbb{R}^d, \forall \boldsymbol{\mu}_0 \in \mathcal{P}(\mathcal{X})$, for some $\chi > 0$, $F_{\boldsymbol{\mu}_0}(\Phi) - \chi I_{\mathrm{d}}$ is positive semi-definite where $F_{\boldsymbol{\mu}_0}(\Phi)$ can be expressed as follows.

$$
F_{\boldsymbol{\mu}_0}(\Phi) \triangleq \mathbb{E}_{(x, \boldsymbol{\mu}, u) \sim \zeta_{\boldsymbol{\mu}_0}^{\Phi}} \left[ \{ \nabla_{\Phi} \pi_{\Phi}(x, \boldsymbol{\mu})(u) \} \right.
$$
$$
\left. \times \{ \nabla_{\Phi} \log \pi_{\Phi}(x, \boldsymbol{\mu})(u) \}^{\mathrm{T}} \right]
$$

**Assumption 6.** $\forall \Phi \in \mathbb{R}^d, \forall \boldsymbol{\mu} \in \mathcal{P}(\mathcal{X}), \forall x \in \mathcal{X}, \forall u \in \mathcal{U}$,

$$
|\nabla_{\Phi} \log \pi_{\Phi}(x, \boldsymbol{\mu})(u)|_1 \leq G
$$

*for some positive constant $G$.*

**Assumption 7.** $\forall \Phi_1, \Phi_2 \in \mathbb{R}^d, \forall \boldsymbol{\mu} \in \mathcal{P}(\mathcal{X}), \forall x \in \mathcal{X}, \forall u \in \mathcal{U}$,

$$
|\nabla_{\Phi_1} \log \pi_{\Phi_1}(x, \boldsymbol{\mu})(u) - \nabla_{\Phi_2} \log \pi_{\Phi_2}(x, \boldsymbol{\mu})(u)|_1
$$
$$
\leq M |\Phi_1 - \Phi_2|_1
$$

*for some positive constant $M$.*

**Assumption 8.** $\forall \Phi \in \mathbb{R}^d, \forall \boldsymbol{\mu}_0 \in \mathcal{P}(\mathcal{X})$,

$$
L_{\zeta_{\boldsymbol{\mu}_0}^{\Phi^*}}(\mathbf{w}_{\Phi}^*, \Phi) \leq \epsilon_{\mathrm{bias}}, \quad \mathbf{w}_{\Phi}^* \triangleq \arg\min_{\mathbf{w} \in \mathbb{R}^d} L_{\zeta_{\boldsymbol{\mu}_0}^{\Phi}}(\mathbf{w}, \Phi)
$$

*where $\Phi^*$ is the parameter of the optimal policy.*

**Lemma 1.** *Let $\{\Phi_j\}_{j=1}^J$ be the sequence of policy parameters obtained from Algorithm 1. If Assumptions $5-8$ hold, then the following inequality holds for some $\eta, \alpha, J, L$,*

$$
v_{\mathrm{MF}}^*(\boldsymbol{\mu}_0) - \frac{1}{J} \sum_{j=1}^J v_{\mathrm{MF}}(\boldsymbol{\mu}_0, \pi_{\Phi_j}) \leq \frac{\sqrt{\epsilon_{\mathrm{bias}}}}{1-\gamma} + \epsilon,
$$

*for arbitrary initial parameter $\Phi_0$ and initial state distribution $\boldsymbol{\mu}_0 \in \mathcal{P}(\mathcal{X})$. The parameter $\epsilon_{\mathrm{bias}}$ is a constant. The sample complexity of Algorithm 1 is $\mathcal{O}(\epsilon^{-3})$.*

The bias $\epsilon_{\mathrm{bias}}$ turns out to be small for rich neural network based policies [Liu et al., 2020]. Intuitively, it indicates the expressive power of the policy class, $\Pi$.

Lemma 1 establishes that Algorithm 1 can approximate the optimal mean-field value function with an error bound of $\epsilon$, and a sample complexity of $\mathcal{O}(\epsilon^{-3})$. Using Theorem 1, we can now state the following result.

**Theorem 2.** *Let $\boldsymbol{x}_0 \triangleq \{x_0^i\}_{i \in [N]}$ be the initial states in an $N$-agent system and $\boldsymbol{\mu}_0$ their associated empirical distribution. Assume that $\{\Phi_j\}_{j=1}^J$ are the policy parameters generated from Algorithm 1, and the set of policies, $\Pi$ satisfies Assumption 3. If Assumptions 1, 2, 4, 5 - 8 are satisfied, then, for any $\epsilon > 0$, the following inequality holds for certain choices of $\eta, \alpha, J, L$*

$$
\left| \sup_{\Phi \in \mathbb{R}^d} v_{\mathrm{MARL}}(\boldsymbol{x}_0, \pi_{\Phi}) - \frac{1}{J} \sum_{j=1}^J v_{\mathrm{MF}}(\boldsymbol{\mu}_0, \pi_{\Phi_j}) \right|
$$
$$
\leq \frac{\sqrt{\epsilon_{\mathrm{bias}}}}{1-\gamma} + C \max\{e, \epsilon\}
$$

*where* $e \triangleq \frac{1}{\sqrt{N}} \left[ \sqrt{|\mathcal{X}|} + \sqrt{|\mathcal{U}|} \right]$

$$\tag{19}$$

*whenever $\gamma S_P < 1$ where $S_P$ is given in Theorem 1. The term, $C$ is a constant and the parameter $\epsilon_{\mathrm{bias}}$ is defined in Lemma 1. The sample complexity of the process is $\mathcal{O}(\epsilon^{-3})$.*

*Proof.* Note that following inequality,

$$\left| \sup_{\Phi \in \mathbb{R}^d} v_{\mathrm{MARL}}(\boldsymbol{x}_0, \pi_\Phi) - \frac{1}{J} \sum_{j=1}^{J} v_{\mathrm{MF}}(\boldsymbol{\mu}_0, \pi_{\Phi_j}) \right|$$

$$\leq \left| \sup_{\Phi \in \mathbb{R}^d} v_{\mathrm{MARL}}(\boldsymbol{x}_0, \pi_\Phi) - v_{\mathrm{MF}}^*(\boldsymbol{\mu}_0) \right|$$

$$+ \left| v_{\mathrm{MF}}^*(\boldsymbol{\mu}_0) - \frac{1}{J} \sum_{j=1}^{J} v_{\mathrm{MF}}(\boldsymbol{\mu}_0, \pi_{\Phi_j}) \right|$$

Using Theorem 1, the first term can be bounded by $C'e$ for some constant $C'$. The second term can be bounded by $\sqrt{\epsilon_{\mathrm{bias}}}/(1-\gamma) + \epsilon$ with a sample complexity of $\mathcal{O}(\epsilon^{-3})$ (Lemma 1). Assigning $C = 2\max\{C', 1\}$, we conclude the result. $\qquad\square$

Theorem 2 guarantees that Algorithm 1 can yield a policy such that its associated value is $\mathcal{O}(\max\{e, \epsilon\})$ error away from the optimal value of the non-uniform MARL problem. Moreover, it also dictates such a policy can be obtained with a sample complexity of $\mathcal{O}(\epsilon^{-3})$.

# 6 EXPERIMENTS

Let the policy sequence that maximizes the mean-field value function $v_{\mathrm{MF}}(\boldsymbol{\mu}_0, \cdot)$ be denoted as $\boldsymbol{\pi}_{\mathrm{MF}}^*$ where $\boldsymbol{\mu}_0$ indicates the empirical distribution of the initial joint state, $\boldsymbol{x}_0^N$. We define the percentage error as follows.

$$\mathrm{error} \triangleq \left| \frac{v_{\mathrm{MARL}}(\boldsymbol{x}_0^N, \boldsymbol{\pi}_{\mathrm{MF}}^*) - v_{\mathrm{MF}}(\boldsymbol{\mu}_0, \boldsymbol{\pi}_{\mathrm{MF}}^*)}{v_{\mathrm{MF}}(\boldsymbol{\mu}_0, \boldsymbol{\pi}_{\mathrm{MF}}^*)} \right| \times 100\%$$

$$(20)$$

We can approximately obtain $\boldsymbol{\pi}_{\mathrm{MF}}^*$ using Algorithm 1. Fig. 1 plots the value of error (defined in (20)) as a function of $N$ for the reward, transition function, and interaction model described in Example 1. The values of various parameters used in this numerical experiment are provided in the description of Fig. 1. Evidently, the error decreases with $N$. Notice that the reward function stated in Example 1 (thereby that is used for generating Fig. 1) is linear in its mean-field distribution argument. In Fig. 2, we exhibit the error as a function of $N$ with the following non-linear reward function.

$$r(x_t^i, u_t^i, \boldsymbol{\mu}_t^{i,N}, \boldsymbol{\nu}_t^{i,N}) = \alpha_R x_t^i - \beta_R (\bar{\boldsymbol{\mu}}_t^{i,N})^\sigma - \lambda_R u_t^i$$

$$(21)$$

The term $\sigma$ is a measure of non-linearity. All other parameters are same as stated in Example 1. Observe that if $\sigma = 1$, the reward function stated above turns out to be identical to the reward function given in Example 1. In Fig. 2a, and 2b we plot error for $\sigma = 1.1, 1.2$ respectively. In both of these scenarios, we see the error to be a decreasing function of $N$.

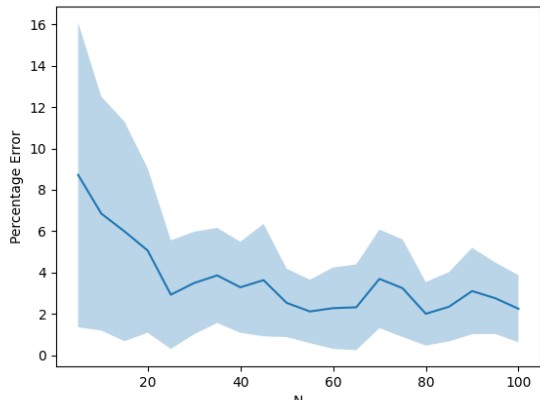

Figure 1: Percentage error (defined by (20)) as a function of $N$. Reward, state transition, and agent interaction matrix are same as stated in Example 1. The bold line and the half-width of the shaded region respectively denote the mean, and the standard deviation values of the error obtained over 25 random seeds. The values of various system parameters used in the experiment are as follows: $K = 5$, $\alpha_R = 1$, $\beta_R = \lambda_R = 0.5$, and $Q = 10$. The hyperparameter values used in Algorithm 1 are as follows: $\alpha = \eta = 10^{-3}$, $J = L = 10^2$. We use a feed forward neural network with a single hidden layer as the policy approximator.

This indicates that although our MFC-based approximation results are theoretically proven for affine rewards only, they empirically hold for non-affine rewards as well.

The codes for generating these results are publicly available at: https://github.com/washim-uddin-mondal/UAI2022

# 7 CONCLUSION

In this article, we consider a multi-agent reinforcement learning (MARL) problem where the interaction between agents is described by a doubly stochastic matrix. We prove that, if the reward function is affine, one can well-approximate this non-uniform MARL problem via an associated Mean-Field Control (MFC) problem. We obtain an upper bound of the approximation error as a function of the number of agents, and also propose a natural policy gradient (NPG) algorithm to solve the MFC problem with polynomial sample complexity. The obvious drawback of our approach is the restriction on the structure of the reward function. Therefore, extension of our techniques to non-affine reward functions is an important future goal.

**Acknowledgements**

W. U. M., and S. V. U. were partially funded by NSF Grant No. 1638311 CRISP Type 2/Collaborative Research: Criti-

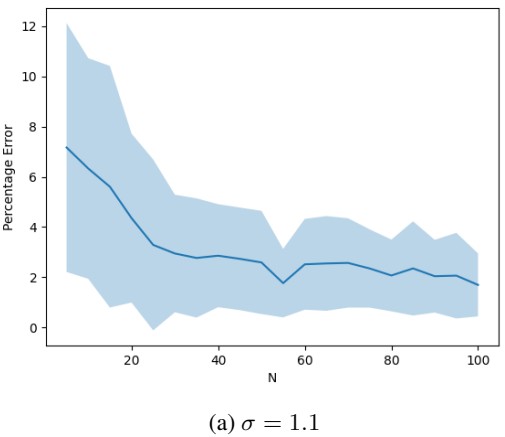 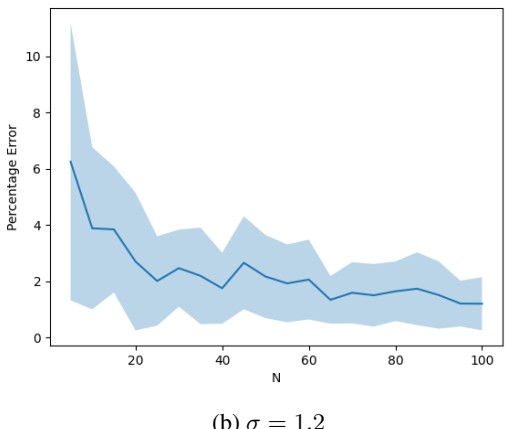

(a) $\sigma = 1.1$            (b) $\sigma = 1.2$

Figure 2: Percentage error when the reward function is given by (21). All other parameters are same as in Fig. 1.

cal Transitions in the Resilience and Recovery of Interdependent Social and Physical Networks.

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
