# OpenReview forum: "Can Mean Field Control (MFC) Approximate Cooperative Multi Agent Reinforcement Learning (MARL) with Non-Uniform Interaction?"
_auai.org/UAI/2022/Conference — UAI 2022 Poster_

### Official Review · Reviewer_fUY4 · 2022-04-03

**Q2(1) Originality/Novelty:** 3
**Q2(2) Significance/Impact:** 3
**Q2(3) Correctness/Technical Quality:** 4
**Q2(6) Clarity Of Writing:** 3
**Q6 Overall Score:** 6
**Q8 Confidence In Your Score:** 4

**Q1 Summary And Contributions:**

This article provides new theoretical results on the use of Mean-Field Control approaches for cooperative MARL problems which are "non-uniform", non-uniformaity being modelled by a doubly stochastic influence matrix W.

**Q2 Assessment Of The Paper:**

More detailed information regarding each of these aspects is given below:

**Q2(4) Quality Of Experiments (Optional):**

2: Fair: The experimental evaluation is weak: important baselines are missing, or the results do not adequately support the main claims.

**Q2(5) Reproducibility:**

3: Good: Key resources (e.g., proofs, code, data) are available and key details (e.g., proofs, experimental setup) are sufficiently well-described for competent researchers to confidently reproduce the main results.

**Q3 Main Strengths:**

The main strengths of the paper are its clear writing, its well-justified contribution and its mathematical strength.

**Q4 Main Weakness:**

The main weaknesses of the paper are its potentially limited contribution, due to the fact that results hold only for affine rewards (but this point is well justified). More important is the complete lack of experimental validation. Experiments showing that good policies can be computed even when affinity does not hold would help convince the reader about the impact of the result.


**Q5 Detailed Comments To The Authors:**

Apart from the strengths and weaknesses pointed out above, something puzzled me and I would like to see this discussed. In the case of non-uniform interaction considered here, is there a reason for the individual policies to be identical, even when r and P are identical? I would think W making the influence of other players heterogeneous would make the optimal policy heterogeneous as well. I really would like to see this discussed in the paper.
Also, I would like to know wheher reward affinity is a necessary condition for MFC to be applicable, or just a sufficient condition.

**Q7 Justification For Your Score:**

I found the paper nice to read and liked the main result, even if of limited applicability. On the other hand, the absence of experimental evaluation is regrettable. I know the algorithm has polynomial sample complexity in the case of affine reward, but an experimental evaluation would help to see whether good results could be obtained for "nearly affine" reward functions, or help discuss the case of heterogeneous policies.
Overall, I think this pape deserves publication.

**Q9 Complying With Reviewing Instructions:**

1: Yes.

---

### Official Review · Reviewer_KgoN · 2022-04-17

**Q2(1) Originality/Novelty:** 3
**Q2(2) Significance/Impact:** 3
**Q2(3) Correctness/Technical Quality:** 3
**Q2(6) Clarity Of Writing:** 3
**Q6 Overall Score:** 6
**Q8 Confidence In Your Score:** 4

**Q1 Summary And Contributions:**

The paper studies the problem of multi-agent RL and investigates the effect of using mean field control when agents interact non-uniformly. The main result is about the difference between the value of a multi-agent policy set for a multi-agent env and its mean field approximation for linear reward functions which decreases at rate $O(1/\sqrt{N})$ and increases with size of state-action space. The authors also provide a convergence result for a NPG style MARL algorithm.

**Q2 Assessment Of The Paper:**

More detailed information regarding each of these aspects is given below:

**Q2(5) Reproducibility:**

4: Excellent: Key resources (e.g., proofs, code, data) are available and key details (e.g., proof sketches, experimental setup) are comprehensively described for competent researchers to confidently and easily reproduce the main results.

**Q3 Main Strengths:**

The authors consider an important aspect of MARL where the interaction between agents is non-uniform and existing results which apply to uniformly interacting agents are no longer applicable. The main result shows that a mean-field approximation of the MARL problem with reward functions linear in the action and state distribution vector of other agents leads to a desired value difference result between the two multi-agent domains. The key requirements for this result is to show that the reward functions are linear and transition dynamics are Lipschitz in the mean-field arguments. Using this, the loss incurred due to the mean field approximation can be bounded at a rate of $O(1/\sqrt{N})$ where $N$ is the number of agents. However, the loss increases with the size of the state-action spaces. This work thus provides an important characterization of when MFC works in MARL settings. For completeness, the authors also provide a MARL algorithm based on NPG and analyze its convergence.

**Q4 Main Weakness:**

Despite the interesting results, I would request the authors to respond to the following questions:

1. The discussion about how $\mu_t$, $\nu_t$ can be thought of as an infinite approximation of $\mu_t^N$ and $\nu^N_t$ is not clear. The notion of $\mu_t$ being a population distribution is misleading without any distributional structure over the $N$ agents. Similarly, $\mu_t^N$, $\nu^N_t$ should depend on the relationships between the finitely many $N$ agents. This discussion is quite unclear to me and am not sure if the intuition is correct.

2. Doubly stochastic assumption: I understand that this is important for the theoretical results, but the authors need to motivate it better and further study if any relaxation is possible. For instance, the assumption clearly holds for symmetric interaction settings.

3. Benchmark policy: In finitely many agent settings, why is enough to compare with the same policy for each agent? The comparator policies can be different for each agent.

**Q5 Detailed Comments To The Authors:**

Refer to the previous question.

**Q7 Justification For Your Score:**

Overall, the work seems solid and addresses an important aspect of MARL. I have raised a few clarity issues in the text and questions relating to the key assumptions made in the work. As the contribution feels significant, I'm leaning towards an accept rating for this work.

**Q9 Complying With Reviewing Instructions:**

1: Yes.

---

### Official Review · Reviewer_G21j · 2022-04-17

**Q2(1) Originality/Novelty:** 3
**Q2(2) Significance/Impact:** 3
**Q2(3) Correctness/Technical Quality:** 3
**Q2(6) Clarity Of Writing:** 3
**Q6 Overall Score:** 5
**Q8 Confidence In Your Score:** 3

**Q1 Summary And Contributions:**

The paper extends the usual mean field control problem to interaction asymmetric case. The paper proves that, if the reward of each agent is an affine function, then one can approximate such a non-uniform MARL problem via its associated MFC problem. And NPG algorithm can then be employed.

**Q2 Assessment Of The Paper:**

More detailed information regarding each of these aspects is given below:

**Q2(4) Quality Of Experiments (Optional):**

3: Good: The experimental evaluation is adequate, and the results convincingly support the main claims.

**Q2(5) Reproducibility:**

3: Good: Key resources (e.g., proofs, code, data) are available and key details (e.g., proofs, experimental setup) are sufficiently well-described for competent researchers to confidently reproduce the main results.

**Q3 Main Strengths:**

Mean field is a common and efficient approximation method. The paper provides solid proof on approximating non uniform MARL problem via MFC problem. Upper bound of the approximation error are also given.

**Q4 Main Weakness:**

Experiment can be done to show the feasibility of the algorithms.

**Q5 Detailed Comments To The Authors:**

As mentioned in the paper the non-uniform interaction is more common in real-world. However, the paper does not give any experiment result as evidence. Is it because the affine reward is somewhat too extreme? As discussed in the introduction, affine reward is used in social or economic networks, is it not that usual MFC problems or MARL problems.
Besides, the different between this work and [Mondal et al., 2021] could be discussed more thoroughly. The identity could be made clearer.


**Q7 Justification For Your Score:**

The extension of MFC to non-uniform is significant but the overall framework should be tested with some experiments.

**Q9 Complying With Reviewing Instructions:**

1: Yes.

---

### Decision · Program_Chairs · 2022-05-15

**Decision:**

Accept (Poster)

**Comment:**

Meta Review: The paper makes an important theoretical contribution to the area of mean field control.  It investigates what happens when agents interact in a non-uniform way while mean field control assumes uniform interaction.  The paper derives some theoretical results under suitable assumptions.  While this is an important contribution that advances the understanding of the applicability of mean field control when agents do not interact in a uniform way, no empirical evaluation is provided.